# The Antimicrobial Activity of *Origanum vulgare* L. Correlated with the Gastrointestinal Perturbation in Patients with Metabolic Syndrome

**DOI:** 10.3390/molecules26020283

**Published:** 2021-01-08

**Authors:** Timea Claudia Ghitea, Amina El-Kharoubi, Mariana Ganea, Erika Bimbo-Szuhai, Tiberiu Sebastian Nemeth, Gabriela Ciavoi, Monica Foghis, Luciana Dobjanschi, Annamaria Pallag, Otilia Micle

**Affiliations:** 1Department of Pharmacy, Faculty of Medicine and Pharmacy, University of Oradea, 1st December Square 10, 410068 Oradea, Romania; timea.ghitea@csud.uoradea.ro (T.C.G.); madafarm2005@yahoo.com (M.G.); snemeth10@gmail.com (T.S.N.); fog1monica@yahoo.com (M.F.); 2Department of Medical Disciplines, Faculty of Medicine and Pharmacy, University of Oradea, 1st December Square 10, 410068 Oradea, Romania; aminaadnan2005@yahoo.com; 3Department of Surgical Disciplines, Faculty of Medicine and Pharmacy, University of Oradea, 1st December Square 10, 410068 Oradea, Romania; bszera@gmail.com; 4Department of Dental Medicines, Faculty of Medicine and Pharmacy, University of Oradea, 1st December Square 10, 410068 Oradea, Romania; gabrielaciavoi@yahoo.com; 5Department of Preclinical Disciplines, Faculty of Medicine and Pharmacy, University of Oradea, 1st December Square 10, 410068 Oradea, Romania; dobjanschil@yahoo.com (L.D.); micleotilia@gmail.com (O.M.)

**Keywords:** metabolic syndrome, *Staphylococcus aureus*, *Escherichia coli*, *Streptococcus pyogenes*, *Origanum vulgare* L.

## Abstract

(1) The metabolic syndrome (MS) promotes acute and chronic infections, due to the pro-inflammatory condition given by TNFα and IL6 or by affecting the microbiota. MS is also correlated with insulin resistance, causing inflammation and infections throughout the organism. (2) The purpose of this study was to track the effect of using the essential oil of *Origanum vulgare* L. (EOO) as an antibacterial treatment, compared to allopathic treatment with antibiotics in MS patients. A group of 106 people with MS was divided into four subgroups: L1—staphylococcal infection group, L2—*Escherichia coli* infection group, L3—streptococcal infection group with EOO treatment, and CG—control group without infection or treatment. (3) EOO is responsible for the antibacterial effect, and reduced minor uncomplicated infections. After a 10-day treatment, intestinal side effects were absent, improving the phase angle. (4) The results suggest that EOO may exhibit an antibacterial effect, similar to the antibiotic treatment, without promoting MS-specific dysbiosis, and it also improves the phase angle in patients, which is used as an index of health and cellular function.

## 1. Introduction

In metabolic syndrome (MS), chronic inflammation is caused by the primary secretion of fat cells of TNFα, and IL6 [1,2]. This is due to the secondary activity of the intestinal mucosa when it comes into contact with foods that produce delayed reactions of type IgG [3]. This mechanism, as shown in Figure 1, activates an immune response from the T cell to the associated lymphoid tissue. Regulatory helper or T cells are associated with anti-inflammatory cytokines (IL10) [4] in their activation or inhibition, and are correlated with both food intolerance reactions and autoimmune diseases in MS. TH3 contributes to the maintenance of homeostasis at the intestinal level. Th1 is involved in cellular response against pathogens, being secreted by macrophages together with pro-inflammatory cytokines. In case of over activation, hypersensitivity occurs to the delayed allergic reaction. Diabetes mellitus, insulin resistance and MS are part of the diseases involved [5]. TH2 is involved in the immune response to an extracellular stimulus, like antibodies, complement proteins and certain antimicrobial peptides. These TH2 cells lead to the secretion of IL4, which in turn stimulates B cells to produce IgE antibodies that activate macrophages, histamine secretion and inflammation, and are correlated with dermatological allergic reactions [6]. The greater the inflammatory reaction, the more devastating the allopathic treatment of the cytotoxic bacteria not only at the pathogen level, in this case *Staphylococcus aureus*, *Streptococcus pyogenes* and *Escherichia coli*, respectively. At the microbiome level, the effects are even more devastating, leading to the worsening of the underlying disease, i.e., MS [7].

The use of broad-spectrum antibiotics leads to the development of bacterial resistance, especially to diseases where wound healing is delayed, and the risk of over-infection is high [8]. The aim of our study was to use the essential oil of *Origanum vulgare* L. (EOO) as an alternative treatment with natural antibiotic, effectively free of significant side effects, which brings great benefits to patients with MS, and doctors offer the possibility of finding diverse treatment variants for minor, uncomplicated acute infections in MS.

Essential oils are used in many natural anti-infectious, antimycotic disinfectant treatments, where they are found alongside EOO and thyme oil, or cloves. The antibacterial activity of natural extracts was studied in a 2020 study, following the expression of NFαB, and the cytokine response of IL6 and TNFα, where significant results were obtained [9]. The antibacterial as well as the antioxidant activity of plants has been highlighted in several studies, both in the case of *Origanum vulgare* L. and other plant species [10]. Our study followed a bacterial infection in the skin, throat and lower digestive tract, treated with EOO in patients with MS, where it was proven effective in reducing the infection, without affecting the intestinal flora.

### Phase Angle

The total mass of the cells was directly proportional to the bio-impact phase or phase angle and is measured by the bioelectric impact cell analyzer. This is an indicator of the amount of electrical charge, based on a ratio between the strength and reaction of the body, which cell membranes can maintain, and are used as an index of health and cellular function. Tracking this parameter is useful as an indicator of catabolism in patients with MS and diabetes [11].

In normal and overweight adults, the phase angle is increased with increased BMI, but there was also a reverse association with a BMI > 40 kg/m^2^. The reference values used are German, where clinical trials were conducted on 15,605 children, adolescents and 21,732 adults, the reference values being established according to BMI, age, sex and health status [12]. In this study, a change in reference values was observed in case of serious diseases, e.g., the cirrhosis of the liver, where the phase angle is inversely proportional to the course of the disease. Several studies are concerned with verifying the prevalence of phase angle use as a predictor of health status, for example in people addicted to alcohol. Alcoholism has a direct link with MS and metabolic diseases due to damage to the mucosa of the small intestine, thus reducing the absorption of nutrients, vitamins and minerals. A pronounced decrease in phase angle was observed in these patients [13].

## 2. Results

### 2.1. Antibacterial Activity of EOO

This study followed the antibacterial activity of EOO on standardized cultures of *Staphylococcus aureus* (ATCC 25923), *Escherichia coli* (ATCC 25922), *Streptococcus pyogenes* (ATCC 19615), which can be traced in Figure 2. On microbial cultures were applied in four micro compresses with four different antibiotics and three different amounts of EOO: 0.2, 0.4 and 0.6 μL.

In the case of *Staphylococcus aureus* (ATCC 25923) cultures, the inhibition zone for azithromycin (15 μg) has a 25 mm diameter, for erythromycin (15 μg), a 32 mm diameter, for levofloxacin (5 μg), a 29 mm diameter and for tetracycline (30 μg), a 30 mm diameter. EOO was used in three different quantities: 0.2, 0.4, and 0.6 μL. A minimum diameter of the inhibition zone of 33 mm was obtained for all quantities used, as shown in Figure 3 and in Table 1.

In the case of *Escherichia coli* (ATCC 25922) cultures, the inhibition zone for azithromycin (15 μg) has a 23 mm diameter, for ampicillin (10 μg), a 24 mm diameter, for levofloxacin (5 μg), a 33 mm diameter and for tetracycline (30 μg), a 29 mm diameter. For EOO, a diameter of 33 mm was obtained for the inhibition zone for all quantities used.

In the case of *Streptococcus pyogenes* (ATCC 19615) cultures, the inhibition zone for ampicillin (10 μg) has a 27 mm diameter, for erythromycin (15 μg), a 30 mm diameter, for levofloxacin (5 μg), a 30 mm diameter, and for tetracycline (30 μg), a 31 mm diameter. For EOO, a minimum diameter of 31 mm was obtained for all quantities used, as shown in Figure 3 and Table 1. The results of this study show the pronounced antibacterial activity of EOO.

Three different amounts of EOO, 0.2, 0.4, and 0.6 μL, were used to determine the minimum inhibitory concentration (MIC). The 0.2 μL amount was the one which determined the diameter higher than that of the most sensitive antibiotic. In conclusion, we reported the antibacterial activity of EOO on bacterial cultures. We obtained the inhibition zone greater than 33 mm for 0.2 μL of EOO, exceeding the inhibition zone of the most sensitive antibiotic. Thus, the use of 0.2 μL of EOO was justified, because for this concentration, the inhibition zone on all bacterial cultures was higher than the inhibition zone of the most sensitive antibiotic.

### 2.2. Study of Bacterial Infection Incidents in Patients with MS

Figure 3 and Table 2 present the results for the relationship between the pre-testing and the post-testing phase in the L1 group. We obtained a Pearson coefficient *r* = −0.268, *p* < 0.001, indicating a statistically significant relationship. At the end of the follow-up period, the number of infections was lower than what it was initially. This relationship results from the negative value of the Pearson coefficient, but also from the value of statistical significance less than 0.05. Regarding the relationship between the pre-testing and post-testing phase in the L2 group, we obtained a Pearson coefficient *r* = −0.208, *p* < 0.05, indicating a statistically significant relationship. At the end of the follow-up period, the number of infections was lower than what it was initially. This relationship results from the negative value of the Pearson coefficient, but also from the value of statistical significance less than 0.05. Following the relationship between the pre-testing and post-testing phase in the L3 group, we obtained a Pearson coefficient *r* = −0.207, *p* < 0.05, indicating a statistically significant relationship. At the end of the follow-up period, the number of infections was lower than what it was initially. This relationship results from the negative value of the Pearson coefficient, but also from the value of statistical significance less than 0.05.

### 2.3. The Status of the Evolution of Gastrointestinal Parameters in Correlation with EOO Therapy

We studied the gastrointestinal parameters using three variables: diarrhea, gastrointestinal pain, flatulence. We used the Pearson statistical analysis.

In the case of diarrhea, as it is shown in Figure 4a, after the treatment of *Staphylococcus aureus* infection, we can observe at the end of the therapeutic period with EOO a decreasing trend. The correlation of the two parameters is also reflected by Pearson statistical analysis, where we obtained a coefficient r = −0.277 (*p* = 0.007). These results indicate a strong relationship between the *Staphylococcus aureus* infection decreases, at the end of the therapeutic period, along with the incidence of diarrhea. In the case of gastrointestinal pain related to staphylococcic infections, shown in Figure 4b, and treated with EOO, a directly proportional correlation can be observed at the end of the treatment (Pearson coefficient *r* = −0.292, *p* = 0.002), i.e., once the staphylococcus infection decreases so does the gastrointestinal pain. This result can be explained with the positive properties of EOO, which do not destroy the intestinal flora, but regulate it. From the correlation between the difference of *Staphylococcus aureus* infection and the difference in flatulence, as outlined in Figure 4c, we obtained a Pearson coefficient *r* = −0.302 (*p* < 0.001), with statistical significance, which shows a strong link, as the two parameters are directly proportionally correlated.

Following the incidence of diarrhea after the treatment of *Escherichia coli* infection, it is shown in Figure 5a, that at the end of the therapeutic period with EOO both *Escherichia coli* infection and diarrhea tend towards “0”. The correlation of the two parameters is reflected by the Pearson statistical analysis. We obtained a coefficient *r* = −0.366 (*p* = 0.001), which indicates a strong relationship between the *E. coli* infection decrease at the end of the therapeutic period and the incidence of diarrhea. EOO, as a natural treatment in *E. coli* infections, has been shown to be effective, with no negative effects on the microbiome, assessing gastrointestinal pain, according to the graph shown in Figure 5b and flatulence, in Figure 5c. Following the statistical analysis, we obtained a Pearson coefficient *r* = −0.174, *p* = 0.06, for gastrointestinal pain and *r* = −0.224 (*p* = 0.01) for flatulence, the last reflecting a strong positive statistical significance.

The incidence of diarrhea after the treatment of *Streptococcus pyogenes* infection is shown in Figure 6a, where at the end of the therapeutic period with EOO, both *Streptococcus pyogenes* infection and diarrhea were tending towards “0”. The correlation between the two parameters is also reflected by Pearson statistical analysis, where we obtained a coefficient *r* = 0.135 (*p* > 0.05), which indicates a positive relationship, but without statistical significance, although all infections have ameliorated with EOO treatment. This result can be explained with insufficient incidence tested (two initial cases). Figure 6b presents the relation between the gastrointestinal pain in the case of *Streptococcus pyogenes* infection, and EOO treatment obtaining a Pearson coefficient of *r* = −0.224, *p* = 0.01 that shows a strongly positive, statistically significant relationship. In the case of flatulence related to the difference in infection with *Streptococcus pyogenes*, as shown in Figure 6c, a directly proportional relationship can be observed, but one that does not reach the significant threshold, the Pearson coefficient being *r* = 0.21, *p* > 0.05.

### 2.4. Changes in the Phase Angle

After the initial and final evaluation of the patients, a statistically significant improvement of the phase angle can be observed. The coefficient of the Student t test for comparing two independent samples (initial and final evaluation) is *t* = −8.108, *p* < 0.001. Significance thresholds indicate an increase in the phase angle for all patients, as seen in Figure 7. For the individual groups analyzed, the following can be observed:-No changes were recorded for the control group (*t* = 1000, *p* = 0.330);-For L1 an improvement *t* = −10.020, *p* < 0.01 is recorded;-For group L2 a significant improvement reflected from the value *t* = −8.099, *p* < 0.01 can be observed;-For group L3 *t* = −6.205, *p* < 0.01, indicating a statistically significant change.

## 3. Discussion

Microbiota plays an important role not only in the management of MS and chronic inflammation but has a direct connection with the immune and neuroendocrine system, as well as with the autonomic and central nervous system [7]. The risks associated with MS of cardiovascular disease, type 2 diabetes, as well as the development of bacterial resistance in infections increase the risk of mortality [14]. Numerous studies follow the evolution of patients with MS, due to the increasing trend of cases, in terms of associated diseases [15], abdominal obesity [16], or the correlation of nutritional status with hepatic steatosis [17]. Some plants promote obesity, resulting from the study published in 2019, but do not influence the development of insulin resistance, when administered for 6 weeks, in rats [18]. There are many studies that highlight the rebalancing effect of the EOO at the intestinal level [18]. Our study had an impact on the treatment of minor infections with EOO instead of allopathic treatment, so it did not worsen flatulence, gastrointestinal pain, or diarrhea, which are signs of destruction of the intestinal microflora. An important role in the defense of the organism against infection has lymphoid tissue associated in the intestinal immunity, which is called innate immunity [19]. Regarding the oxidative stress in metabolic syndrome and diabetes, we can also intervene with the help of plants [20]. In 2016, Andersen et al. highlighted the effect of obesity on the immune response to pathogens, as well as a disruption of the function of associated lymphoid tissue due to fat tissue [21]. Reduced neutrophil activity due to MS and obesity increases susceptibility to *Staphylococcus aureus* infections [22,23,24]. We obtained a comparable result in treatment with EOO instead of antibiotics, in minor infections with Staphylococcus sp. It allowed the normal function of the immune system in the intestinal tract. MS being correlated with hyper coagulation has a decisive role in the management of infections with *Streptococcus pyogenes*, through the formation of fibrin [25]. Infections with *Streptococcus pyogenes* are in smaller numbers, which is reflected in the number of patients included in the study with this infection. However, the results are encouraging, and allow us to propose longer studies, regarding a greater group of patients. *E. coli* infections have been a major problem, especially in the last decade. The persons concerned are those who have a lower immunity [26], including those with MS, due to impaired lymphoid tissue. Treating infections in the early stages with natural antibiotics, and in the case of L2, with *E. coli* infections, these have proven effective, also having lower cytotoxicity than allopathic treatment.

EOO is used for its multiple beneficial effects on human health [21], but also as a prophylactic treatment with high potential [27] in the agro-food industry, replacing the antibiotics used in birds with EOO. Studies in 2019, following infections with *Staphylococcus aureus*, the resistant form, observed an inhibitory activity of EOO. In patients with MS, the effective minimum dose reduces side effects, and interactions with drugs. Carvacrol, being one of the main components of EOO, activates the potential transient receptor (TRP) channels TRPA1 and TRPV3 [28], and produces endothelial vasodilation. This cardioprotective effect increases the value of use in people with MS. EOO in *Streptococcus pyogenes* infections has been shown to be effective in treating pharyngitis [29]. Effective minimum concentrations for the destruction of the bacterial wall are followed in several follicles of pathogens, as an activity of the essential oils [30]. The treatment applied to patients with minor infections, was tracked and evaluated, not only from a symptomatological point of view, but also from its correlation with the side effects of antibiotics in people with MS [31]. Alternative treatments from the class of cefmetazole cephalosporins have been studied, phosphomycin being compared to ciprofloxacin [32], but the side effects of antibiotics increase along with the spectrum of activity and bacterial resistance. Urinary and vaginal infections caused by *Escherichia coli* were successfully treated with EOO [33]. In patients with MS in the current study, EOO inhibited the development of *Escherichia coli*, improving gastrointestinal health, which cannot be obtained with allopathic treatment [34]. In the case of the correlation of gastrointestinal problems with the treatment for *E. coli* infections, a strong, statistically significant correlation was found, which can also be explained by the large number of participants in the L2 group. The improved but statistically insignificant results in L3 can be explained with the small number of participants in the study, but the positive results encourage us to expand the study in the future.

The phase angle, being a component of health tracking during convalescence, is an important predictor [35], but also an indicator of catabolism in type 2 diabetes [9]. MS is correlated with insulin resistance [36] in obese people. In our study, we can follow the positive evolution of the phase angle in all three research groups. This determination of the phase angle has a special importance, in the pursuit of cellular functions. Firstly, because it can be determined non-invasively, and secondly, we can observe cellular decline before reaching a critical threshold.

## 4. Materials and Methods

### 4.1. Ethics

This study was carried out in agreement with the research ethics commission within the Faculty of Medicine and Pharmacy, University of Oradea, no. 12/01.04.2019. Before enrolling a patient in the study, the researcher explained the study design to each candidate patient using a brochure and an informed consent form. Written informed consent was given by each patient prior to study participation.

### 4.2. The EOO Obtaining

The medicinal plant species *Origanum vulgare* L. (family Lamiaceae) were used in our study. The plant samples came from unpolluted areas of cultivated flora in Oradea (Oradea, Bihor County, Romania) and were carefully collected and selected in 2018. A specimen of the species is kept in the Oradea Pharmacy Herbarium, code: UOP 05045, (Department of Pharmacy, University of Oradea, Faculty of Medicine and Pharmacy.

The volatile oil was obtained by the hydro-distillation method. The process consists of the following operations: the dried plant material (25 g) is brought together with the water in the distillation flask, the bottom of the vessel is heated directly at the heat source, and the mixture of oil and water vapor is collected by condensation in a system for cooling. The distillate is then passed into a collecting vessel. Phase separation was observed, from where the essential oil (4–5 mL) was collected by settling, as shown in Figure 8 [37].

EOO, obtained from the *Origanum vulgare* L., by hydro-distillation, was diluted with Oleum Helianti 1:1, to reduce the cytotoxicity to the lowest effective level [38].

The verification of the percentage composition of the EOO was performed with a gas chromatograph coupled with mass spectrometry. The ability and high resolution to provide precise and accurate qualitative and quantitative data established gas-chromatography (GC) coupled with mass spectrometry (MS), i.e., GC–MS analyses as a valuable means for the taxonomic research of plants. The relative amounts of individual components of the volatiles of the two samples were expressed as percentages of the peak area relative to the total peak area. Relative percentage amounts were calculated from the TIC by the computer.

The neutral molecules elute from the analytical column and are ionized in the ion source to produce molecular ions which can degrade into fragment ions. The fragment and molecular ions are then separated in the mass analyzer by their mass: charge (*m*/*z*) ratio and are detected. This method is used to identify unknown analytes and to determine the structural and chemical properties of molecules, as well as the chromatogram that can be used for qualitative and quantitative analysis.

The GC–MS analysis of various organic crude extracts isolated from leaves of *Origanum vulgare* L. was performed using a Thermo GC–MS (Model Trace 1310 ISQ 7000) equipped with an HP−5MS capillary column (30 m length × 0.32 mm internal diameter × 0.25 µm film thickness). GC–MS spectroscopic detection, an electron ionization system with an ionization energy of 70 eV was used. Helium was used as a carrier gas at 30 cm s^−1^, and the injection volume was 1 µL. The mass transfer line and injector temperature were set at 220 °C and 290 °C, respectively. The oven temperature was programmed at 45 °C for 1 min, raised to 250 °C at 5 °C min^−1^, and maintained at 250 °C for 5 min. Diluted samples (1/100, *v*/*v*, in dichloromethane) of 1 µL was injected in the split mode with a split ratio of 120:1. The relative percentage of the chemical constituents in the dried extracts of *Origanum vulgare* L. was expressed as the percentage by peak area normalization.

Qualitative and semi-quantitative analysis was applied to the oil obtained and was calculated the retention indices Kovats. Retention indices also find applications in the characterization of the selectivity of stationary phases, in structural analysis, and in studies of physico-chemical properties of stationary phases and analytes. Relationships between Kovats indices and thermodynamic properties are used for the determination of vapor pressures, entropies of adsorption, enthalpies and the vaporization of different analytes. Following the analysis, the result was obtained, which is detailed in Table 3. According to the results presented in Table 3, the presence of carvacrol of 29.22% is observed as the main substance. 

Kovats retention indices (***RI***) formula IS:$$\mathrm{RI}=100(n)\;+\;100(m-n)\frac{tri-trn}{trm-trn}$$
where:***RI*** = retention indices of “***i***”;***i*** = constituent of essential oil that is being analyzed;***n*** = carbon number of the alkane which elutes before “***i***”;***m*** = number of carbons of the alkane which elutes after “***i***”;***tri*** = retention time of “***i***”;***trn*** = retention time of the alkane which elutes before “***i***”;***trm*** = retention time of the alkane which elutes after “***i***”.

### 4.3. Antimicrobial Activity

The antimicrobial activity of the essential oil of *Origanum vulgare* L. was evaluated by the disc diffusion method using the standard methodology [39] and by determining the minimum inhibitory concentrations (MICs). In our study, we used the following bacterial cultures, reference microbial strains: *Staphylococcus aureus* (ATCC 25923), *Streptococcus pyogenes* (ATCC 19615), and *Escherichia coli* (ATCC 25922), (Microbiologicals, St. Cloud, MI, USA).

Kirby–Bauer Disk Diffusion test: for *Staphylococcus aureus* and *Escherichia coli*, Mueller–Hinton Agar (Oxoid) was used, and for *Streptococcus pyogenes,* the medium Mueller–Hinton 2 agar + 5% sheep blood (BioMerieux, Marcy l‘Etoile, France) was used.
The culture medium is a solid Mueller–Hinton agar medium in a uniform layer of 4 mm (stored at 4 °C, for a maximum of 7 days) sown with a bacterial culture;The discs with AB are placed on the surface of the medium;AB will diffuse into the environment, showing a constant decrease in the concentration gradient from the edge of the disc to the periphery;After incubation, two distinct areas are delimited: an area where bacterial growth is inhibited by AB concentration and a growth area where AB concentration is too low to inhibit growth;The larger the diameter of the inhibition zone, the more sensitive the bacteria (MIC is smaller).

Each strain was placed on the appropriate culture medium with a sterile cotton swab and the flat swab, and the plates were dried for 10–15 min.

The extracted oil was diluted 1:1 with Oleum Helianti (sunflower oil) (HiMedia Laboratories, Einhausen Germany) and with a micropipette, 0.2, 0.4, and 0.6 μL, of EOO mixture were added, placed on inoculated plates.

After overnight incubation at 37 °C, the diameters of the inhibition zone were measured in millimeters. The microtablets with standard antibiotics were used for each strain as positive controls: for *Staphylococcus aureus*, azithromycin (15 μg), erythromycin (15 μg), levofloxacin (5 μg) and tetracycline (30 μg) were used; for *Escherichia coli*, azithromycin (15 μg), ampicillin (10 μg) levofloxacin (5 μg) and tetracycline (30 μg) were used; and for *Streptococcus pyogenes*, ampicillin (10 μg), erythromycin (15 μg), levofloxacin (5 μg) and tetracycline (30 μg) were used. Each test was performed in triplicate and the mean values were selected.

Minimum inhibitory concentration (MIC) is defined as the lowest concentration of a drug that will inhibit the natural growth of an organism after overnight incubation. Three different amounts of EOO—0.2, 0.4, 0.6 μL—were used to determine the minimum inhibitory concentration.

We evaluated the largest diameter of the most sensitive antibiotic in order to stabilize the lowest efficiency of EOO with MIC [40].

### 4.4. Clinical Trials

The study was performed on 106 patients, aged over 18 years, the average age being 37.42 years (±14.12), divided into 4 groups:-CG = patients in control group without infection (*n* = 20);-L1 = patients with Staphylococcus infection (*n* = 32);-L2 = patients with Escherichia coli infection (*n* = 51);-L3 = patients with Streptococcus infection (*n* = 2).

All patients were diagnosed with MS without CG infections, and without minor, uncomplicated infections with *Staphylococcus aureus* (L1), *Escherichia coli* (L2), *Streptococcus pyogenes* (L3), which were the inclusion criterion.

The exclusion criteria were people who presented infections by other strains than the studied three, and/or who presented other major or complicated infections.

All patients included in the study agreed with the natural treatment without allopathic treatment. They received 0.8 mL (2 drops) EOO twice a day, for ten days, as treatment.

### 4.5. Enrolled Patients and Anthropometry

We conducted a cross-sectional study on MS patients that were enrolled in a cohort study, to obtain the clinical characteristics of MS patients and adapted the diet therapy for each patient. The clinical evaluation was performed with the Tanita MC780MA bioelectric impedance body analyzer (Tokyo, Japan) [41], and the results were evaluated using GMON 3.4.1 medical software (Chemnitz, Germany). BIA type body analyzers are devices accepted by the WPHNA (World Public Health Nutrition Association) and were used to determine body composition with high accuracy. The margin of error was 0.1 kg. We followed the evaluation of the affinity for diet therapy with the non-invasive medical device Cnoga MTX(Or-Akiva, Israel), which helped to follow the changes of the clinical parameters as a whole, checking the oxygen saturation, the blood pressure, and the blood pH.

We followed the variations in the four independent groups, depending on sex, age, rural/urban environment, clinical parameters such as BAM, weight status, fat mass, visceral fat, hydration status (ECW/TBW), sarcophagi index, phase angle, basal metabolism (BMR), and pH, followed by MS.

The completion of the evaluation sheets of the patients included in the research study took place at the beginning and at the end of the study period by the participants.

### 4.6. Tracking Metabolic Parameters, Phase Angle

The total mass of the cells was directly proportional to the bioimpedance phase or phase angle and is measured by the bioelectric impedance body analyzer. Patients were evaluated at the beginning and the end of the treatment with this device. With bare feet, they climbed on the base of the scale and held two electrodes in their hands. The measurements lasted up to 10 s after which the results were evaluated. According to the results, we could see their improvement or deterioration, and when analyzed in the statistical program SPSS 22 (New York, NY, USA), we could follow both the final result compared to the initial values and the possible correlations between health (measured by phase angle) and the healing of infections in patients with MS.

### 4.7. Tracking the Status of the Evolution of Gastrointestinal Parameters

To assess gastrointestinal status for diarrhea, flatulence, and gastrointestinal pain, patients completed an evaluation form at the beginning and at the end of the study period. On the sheet with “0”, the symptoms absence was noted and with “1” their presence was noted.

#### 4.7.1. Tracking Diarrhea Status

Diarrhea was followed during treatment as a sign of dysbiosis, which is correlated with the use of antibiotics; it was noted in patients if they had 3 consecutive stools of soft consistency. This was followed throughout the treatment with EOO.

#### 4.7.2. Tracking the Flatulence

Flatulence, present as an associated problem of MS, was monitored if it was different from that before the start of treatment, or if it occurred during this period. We also followed the flatulence of MS by completing the evaluation form at the beginning and at the end of the therapy, marking with “0” the absence of symptoms and with “1” their presence.

#### 4.7.3. Tracking Gastrointestinal Pain

Gastrointestinal pain was evaluated with adapted scales (visual analogue scale (VAS), numerical rating scale (NRS), and verbal rating scale (VRS) [42]) during treatment, as a sign of dysbiosis, which is correlated with the use of antibiotics. During treatment with EOO, the study sheet was completed at the beginning and at the end of therapy with “0” marking symptomatic absence and “1” marking the presence of pain.

### 4.8. Statistical Analysis

The exit rate of the people in the study was 0.00%, so we obtained, with the Chi-Square test, the lots with statistical significance Asymp. Sig. < 0.05. SPSS Statistics 22 was used to perform the statistical analysis [43]. All parameters’ mean values, frequency intervals and standard deviations, as well as tests of statistical significance, were calculated using the Student t test. The lots were distributed similarly to the normal one, using assumptions involving numerical data.

The Bravais–Pearson correlation coefficient was used to calculate an independent indicator of the units of measurement of the two variables. The value of *p* < 0.05 was assigned to ANOVA statistical significance, as well as *p* < 0.01 indicating high level statistical significance. Bonferroni post hoc analysis was used to analyze the differences between groups, as an additional analysis of subgroups.

## 5. Conclusions

EOO is active on bacteria such as *Staphylococcus aureus*, *Escherichia coli* and *Streptococcus aureus*, the minimum concentration of inhibition having no cytotoxic effects, so it can be recommended for minor infections in patients with MS.

With the EOO natural treatment, the results of patients with MS and minor infections with *Staphylococcus aureus*, *Streptococcus pyogenes* and *Escherichia coli* improved, without significant damage to the intestinal microbiome. Assessing diarrhea, gastrointestinal pain and flatulence, we found that almost every parameter improved in a statistically significant way. With the EOO treatment, in the case of minor infections in patients with MS, the phase has also improved, indicating an increase in health and an improvement in cellular functions. However, further studies, longer study periods and a larger number of patients are needed in order to determine the exact mechanisms underlying the effects of EOO.

## Figures and Tables

**Figure 1 molecules-26-00283-f001:**
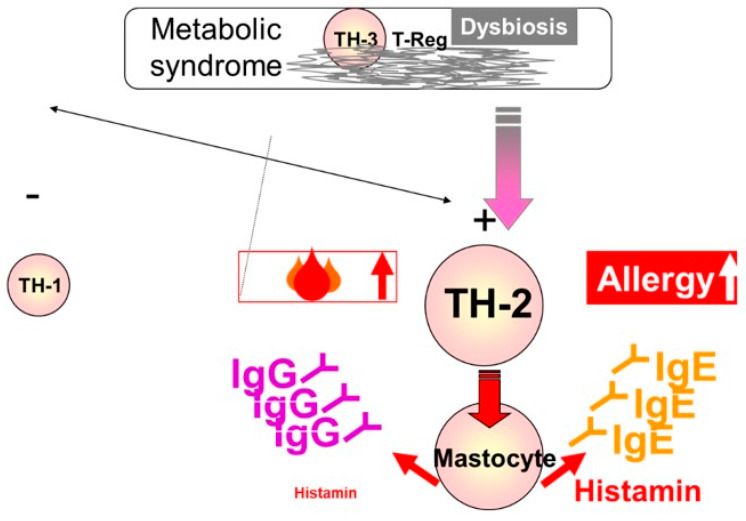
T cell immune response mechanism.

**Figure 2 molecules-26-00283-f002:**
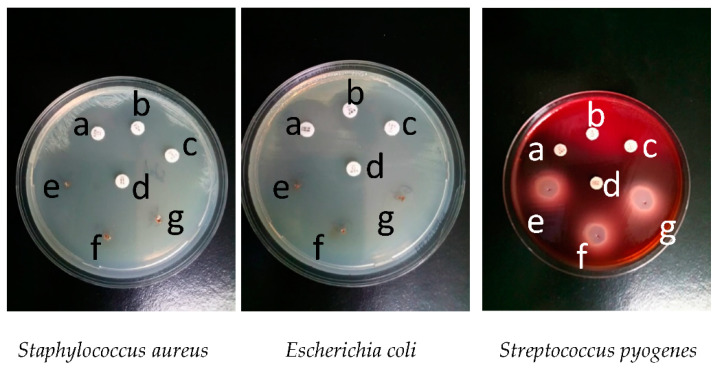
Antibiograms on the cultures of *S. aureus* (a = azithromycin 15 μg, b = erythromycin 15 μg, c = levofloxacin 5 μg, d = tetracycline 30 μg, e = *Origanum vulgare* L. (EOO) 0.2 μL, f = EOO 0.4 μL, g = EOO 0.6 μL), *E. coli* (a = tetracycline 30 μg, b = ampicillin 10 μg, c = levofloxacin 5 μg, d = azithromycin 15 μg, e = EOO 0.2 μL, f = EOO 0.4 μL, g = EOO 0.6 μL) and *S. pyogenes* (a = ampicillin 10 μg, b = erythromycin 15 μg, c = levofloxacin 5 μg, d = tetracycline 30 μg, e = EOO 0.2 μL, f = EOO 0.4 μL, g = EOO 0.6 μL).

**Figure 3 molecules-26-00283-f003:**
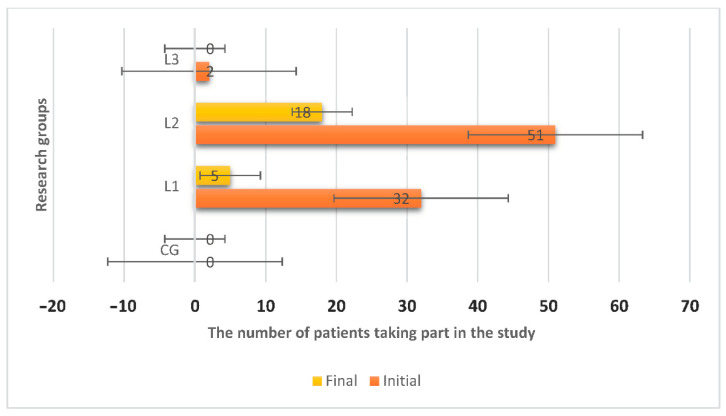
Evolution of the number of infections in the 4 research groups (CG = 20 patients with 0 initial infection, L1 = 32 patients with 32 initial infections, L2 = 51 patients with 51 initial infections, L3 = 2 patients with 2 initial infection).

**Figure 4 molecules-26-00283-f004:**
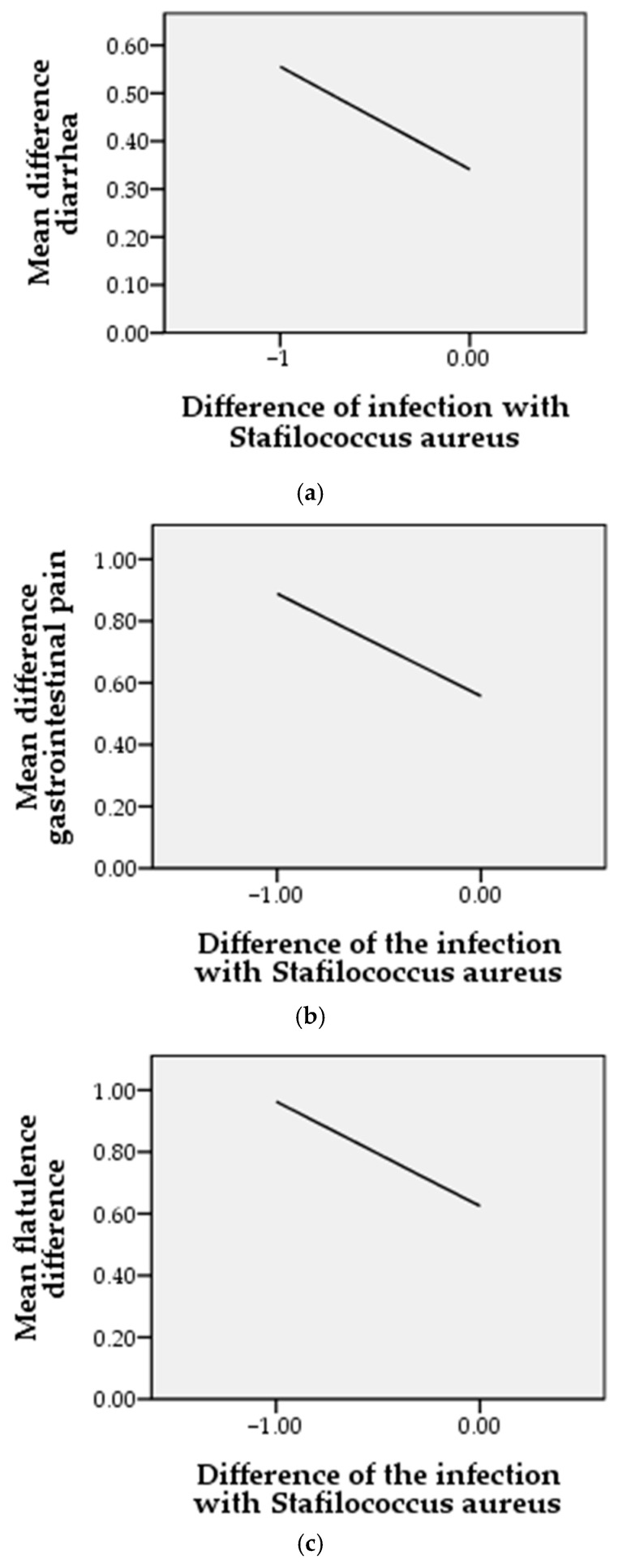
Correlation between gastrointestinal incident and *Staphylococcus aureus* infection at the end of the treatment period where “1” are noted as the presence and “0” are noted as absence of conditions. (**a**) = mean difference diarrhea, (**b**) = mean difference gastrointestinal pain, (**c**) = mean difference flatulence.

**Figure 5 molecules-26-00283-f005:**
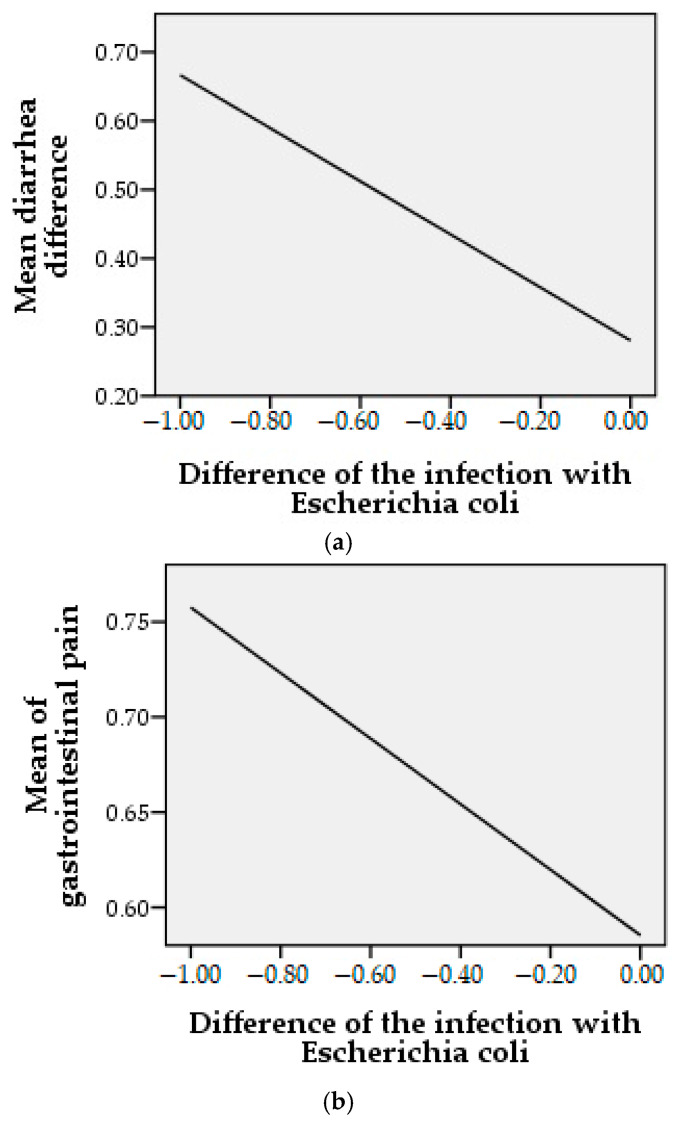
Correlation between the incidence of gastrointestinal symptoms and *Escherichia coli* infection at the end of the treatment period where “1” are noted as the presence and “0” are noted as absence of conditions.; (**a**) = mean difference diarrhea, (**b**) = mean difference gastrointestinal pain, (**c**) = mean difference flatulence.

**Figure 6 molecules-26-00283-f006:**
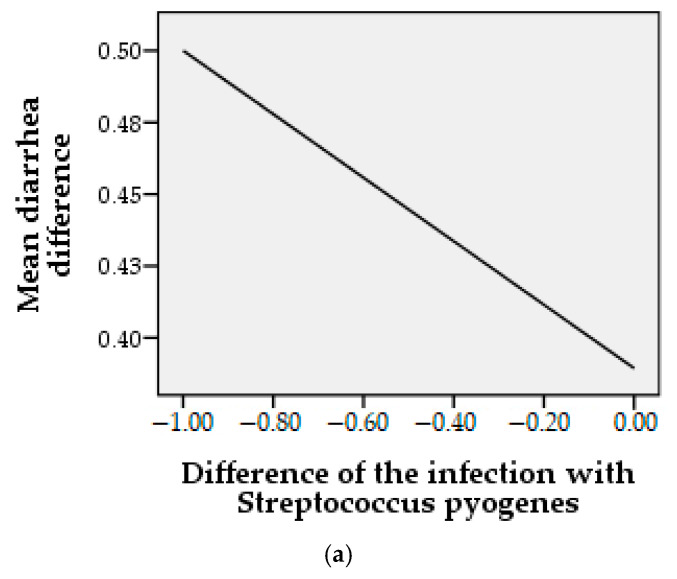
Correlation on gastrointestinal incidence and infection with *Streptococcus pyogenes* at the end of the treatment period where “1” are noted as the presence and “0” are noted as absence of conditions.; (**a**) = mean difference diarrhea, (**b**) = mean difference gastrointestinal pain, (**c**) = mean difference flatulence.

**Figure 7 molecules-26-00283-f007:**
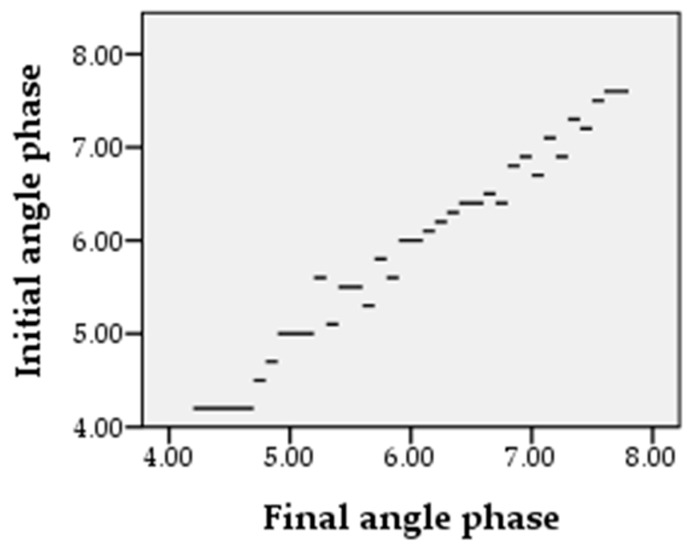
Evolution of the phase angle from the beginning to the end of the treatment period at L1, L2 and L3 measured in degrees (50 kHz).

**Figure 8 molecules-26-00283-f008:**
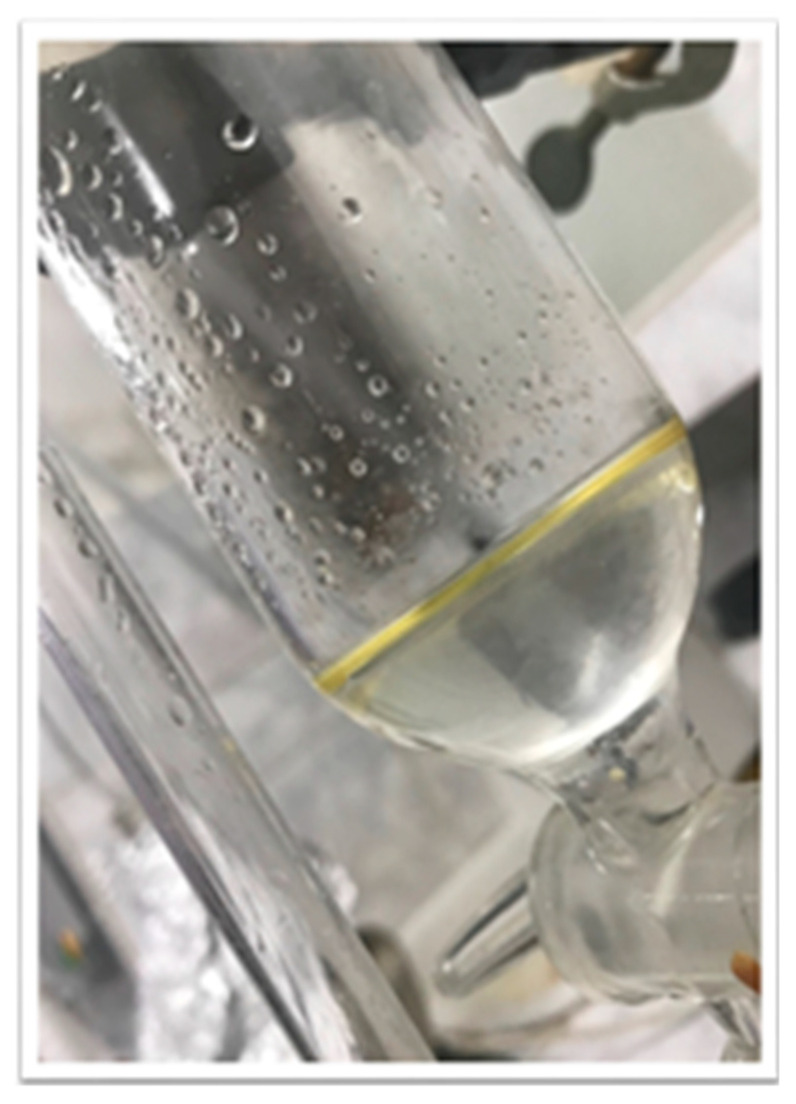
EOO obtained with hydro-distillation extraction.

**Table 1 molecules-26-00283-t001:** Diameter obtained from the EOO antibiotic in the 3 research groups (in millimeters).

*Staphylococcus aureus*		*Escherichia coli*		*Streptococcus pyogenes*	
Antibiotics	Diameter (mm)	Antibiotics	Diameter (mm)	Antibiotics	Diameter (mm)
Azithromycin	25	Azithromycin	23	Ampicillin	27
Erythromycin	32	Ampicillin	24	Erythromycin	30
Levofloxacin	29	Levofloxacin	33	Levofloxacin	30
Tetracycline	30	Tetracycline	29	Tetracycline	31
EOO 0.2 μL	33	EOO 0.2 μL	34	EOO 0.2 μL	32
EOO 0.4 μL	35	EOO 0.4 μL	35	EOO 0.4 μL	33
EOO 0.6 μL	36	EOO 0.6 μL	36	EOO 0.6 μL	34

**Table 2 molecules-26-00283-t002:** Pearson correlation on the difference between the pathogen lots and the research variables. Statistical significance.

Research Variables		Difference in Infection with *Staphylococcus aureus*	Difference in Infection with *Escherichia coli*	Difference in Infection with *Streptococcus pyogenes*
Pearson coefficient	Group 106 patients			
	Diarrhea	−0.277 **	−0.366 **	0.132
	Gastrointestinal pain	−0.292 **	−0.174 *	−0.224 **
	Flatulence	−0.302 **	−0.224 **	0.021

* Pearson coefficient *r* < 0.40 with statistical significance *p* < 0.05. ** Pearson coefficient *r* < 0.40 with statistical significance *p* < 0.01.

**Table 3 molecules-26-00283-t003:** Gas chromatograph report (g × 100 g^−1^ DW) for the extract of EOO in percentage composition.

Compound Name	Retention Time	Relative Area %	Kovats Retention Indices (***RI***)
α-Pinene	6.074	0.23	713
1-Octen-3-ol	6.710	0.77	966
Bicyclo[2.2.1]heptan, 2,2-dimethyl-3-methylene-, (1S)-	6.323	0.16	1055
endo-Borneol	9.737	3.39	1214
2-Furanmethanol, 5-eteniltetrahydro- α,α,5-trimethyl-, cis-	8.248	0.23	1239
2-Methyl-5-(propan-2-yliden)cyclohexan-1,4-diol	11.877	0.98	1244
*o*-Cymene	7.489	0.44	1251
*p*-Cymene-2-ol-methyl-ether	10.790	0.18	1256
cis-Linalool oxide	8.496	0.16	1265
Thymol	11.445	3.62	1268
Carvacrol	11.670	29.22	1292
Carvone	11.118	0.61	1299
(R)-lavandulyl acetate	12.414	0.17	1341
4,4-Dimethylpent-2-enal	11.969	0.19	1369
Terpineol	10.210	1.06	1381
R-(-)-*p*-Menth-l-en-4-ol	9.890	2.89	1409
α-Cadinol	15.997	0.02	1500
3-Octanol	6.969	0.19	1601
Eucaliptol	7.622	0.40	1670
.tau.-Cadinol	15.989	0.57	1804
(-)-Spatulenol	15.302	0.46	1829
Linalool	8.659	11.18	1861
*o*-lzopropilphenethol	12.033	0.27	1895
Phenol, 2-methil-5-(1-methyletil)-, acetate	12.574	0.32	1953
Caryophyllene	13.356	1.44	1962
β-Bisabolene	14.349	2.65	2264
2,7-Dimethyloctadiin-3,5-diol-2,7	13.734	0.52	2288
*S*-lndacen-l-(2H)-ona, 3,5,6,7-tetrahydro-3,3,5,5-tetramethyl-8-(3-methilbutil)	21.508	0.20	3101
3-Trifluoroacetate ester pregnenolone	21.107	0.46	3162
Caryophyllene oxide	15.397	1.01	3669
Geranyl acetate	12.656	0.23	4360

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
