# Peer review of "The Antimicrobial Activity of *Origanum vulgare* L. Correlated with the Gastrointestinal Perturbation in Patients with Metabolic Syndrome"

_molecules, 2021, doi:10.3390/molecules26020283_

Round 1
Reviewer 1 Report
The paper entitled "Correlation of treatment with the essential oil of Origanum vulgare L. of minor bacterial infections with the incidence of dysbiosis in patients with Metabolic Syndrome" showed the antibacterial activity of essential oil of Origanum vulgare L. Overall, the study presented by the authors has good potential. However, an extensive editing of English language is necessary. Authors do not use academic English and in many paragraphs is very difficult to understand what they want to communicate. Authors used “essential oil” of Origanum vulgare (EOO) in their experiment. The essential oils are normally composed by lipophile compounds. However, authors obtained the EOO by hydro distillation in Soxhlet. Did authors used only water or other organic solvent? If they used only water, did they observe phases separation after extraction? If yes, how did they manage the right amount for oral administration in patients? While if authors used organic solvents mixed with water, did they evaluate toxicity before oral administration?
The presented version has lack of information. Results are not presented clearly making the readers difficult to follow even small sentences.
Please see below for detailed comments.
Title: The title can be improved. A good title is short, concise and informative.
Abstract: Do not add references in the Abstract. Move the cited reference in the Introduction.
Lines 22-25: Please, rephrase it.
Line 26-28: Please use the "italic" for latin name as "Origanum vulgare", Staphylococcus aureus, Escherichia coli and Streptococcus pyogenes. Please, apply this change all over the paper.
Line 25-29: This paragraph is not clear. Please rephrase it. Try to split in more sentences.
Line 35: Add a dot after dysbiosis and start a new sentence. Rephrase "in patients also improving the phase angle, used as an index of health and cellular function".
Keywords: Do not use as keywords the same words that are already present in the Title, such as Origanum vulgare, dysbiosis and essential oil.
Results:
Lines 93-95: please delete the double comma.
Figure 2: Please describe better the legend of the figure. What are the disks indicated in the figure 2? Antibiotics? EOO? Control?
Table 1: What does mean a diameter higher than 33? Can also be 100? Please indicate the precise diameters. In the table change the word “Oregano” with “Origan”
Line 131: Please delete the capital letter “The”
Figure 3: this figure is not well presented. Please add axis title, for example what numbers represent. Remove the Title of the figure and try to add more information in the description of the figure.
Line 139: please change the word “variabile” with "variables".
Materials and methods:
Whic solvent did you use for the extraction? What is the ratio between solvent and organic matter? How much solvent and how much Origanum vulgare did you use?
Line 288: Please add a reference here.
Line 298: Please rephrase the sentence “the culture medium is agar is a solid medium Mueller Hinton…”
Line 306: please rephrase, there is no verb in the sentence, there is no end, there is no subject. This is not academic English.
Line 316: correct the ampicillin (10
Line 325: Rename the paragraph for example - Orgnaization of the study groups. - Clinical trials
Lines 329-331: Please substitute with:
- L1 = patients with Staphylococcus infection (n=32)
- L2 = patients with Escherichia coli infection (n=51)
- L3 = patients with Streptococcus infection (n=2).
How did you do statistic in L3 group with only 2 patients?
Line 358: in "their" hands.
Line 366: Please rephrase the sentence.
Line 377: Gastrointestinal pain WAS EVALUATED....
Author Response
Response to Reviewer 1
Firstly, we, the authors of the present manuscript wish to thank you for thoughtful commentary you have provided to improve the quality of the paper. We are very grateful for the time and effort you have devoted to this task. We have extensively revised our manuscript according to the recommendations. All changes in the text and the new figures that we have redesigned are highlighted. Please, see the point-by-point answers to your comments below. The whole manuscript was extensively revised and corrections were made throughout.
Comment 1:
The paper entitled "Correlation of treatment with the essential oil of Origanum vulgare L. of minor bacterial infections with the incidence of dysbiosis in patients with Metabolic Syndrome" showed the antibacterial activity of essential oil of Origanum vulgare L. Overall, the study presented by the authors has good potential. However, an extensive editing of English language is necessary. Authors do not use academic English and in many paragraphs is very difficult to understand what they want to communicate. Authors used “essential oil” of Origanum vulgare (EOO) in their experiment. The essential oils are normally composed by lipophile compounds. However, authors obtained the EOO by hydro distillation in Soxhlet. Did authors used only water or other organic solvent? If they used only water, did they observe phases separation after extraction? If yes, how did they manage the right amount for oral administration in patients? While if authors used organic solvents mixed with water, did they evaluate toxicity before oral administration?
The presented version has lack of information. Results are not presented clearly making the readers difficult to follow even small sentences.
Answer 1
Thank you very much for the observation. The volatile oil was obtained by the hydrodistillation method. There was confusion with the extraction of the fatty oil. The extraction method is detailed between lines 297-302, and we also include the figures of the extraction phases.
The EOO obtained was analyzed with a GC-MS gas chromatograph. The result is shown in Table 3, in line 316. Carvacrol was obtained as the main substance, substance being present 29.22%.
Table 3. Gas chromatograph report for standard and extract of EOO in percentage composition
|
Compound name |
Standard area % |
Extract area % |
Compound name |
Standard area % |
Extract area % |
|
Methyl-alpha-methyl butyrate |
0.05 |
0.06 |
trans-beta-O cymene |
0.05 |
0.18 |
|
alpha-Thujene |
0.62 |
0.52 |
gamma-Terpinene |
4.02 |
1.06 |
|
alpha-Pinene |
0.44 |
0.23 |
cis-Sabinene hydrate |
0.26 |
0.16 |
|
Thujadiene |
0.01 |
|
Terpinolene |
0.11 |
0.09 |
|
Camphene |
0.15 |
0.01 |
T-Linalool oxide (furanoid) |
0.03 |
0.16 |
|
1-Octen-3-ol |
0.26 |
0.77 |
para-Cymenene |
0.02 |
0.18 |
|
Myrcene |
1.34 |
|
Linalool |
8.99 |
11.18 |
|
3-Octanol |
0.01 |
0.19 |
trans-Sabinene hydrate |
0.04 |
0.01 |
|
alpha-Phellandrene |
0.17 |
0.02 |
Hotrienol |
0.13 |
0.01 |
|
delta-3-Carene |
0.06 |
0.03 |
cis-para-Menth-2-en-1-ol |
0.04 |
0.02 |
|
alpha-Terpinene |
0.94 |
1.09 |
alpha-Campholenal |
0.03 |
0.01 |
|
para-Cymene |
2.92 |
0.18 |
trans-Pinocarveol |
0.02 |
0.02 |
|
Limonene |
0.17 |
1.12 |
trans-para-Menth-2-en-1-ol |
0.03 |
2.89 |
|
beta-Phellandrene |
0.18 |
0.25 |
Borneol |
0.7 |
3.39 |
|
1,8-Cineole |
0.06 |
0.02 |
Terpinen-4-ol |
0.69 |
1.06 |
|
cis-beta-Ocimene |
0.03 |
0.01 |
para-Cymen-8-ol |
0.03 |
0.18 |
|
alpha-Terpineol |
0.25 |
1.06 |
alpha-Bourbonene |
0.02 |
0.00 |
|
cis-Dihydro carvone |
0.03 |
0.01 |
beta-Caryophyllene |
0.93 |
1.44 |
|
trans-Chrysanthenyl acetate |
0.02 |
0.01 |
trans-alpha-Bergamotene |
0.02 |
0.01 |
|
Carvacrol Methyl ether |
0.05 |
|
Aromadendrene |
0.18 |
0.01 |
|
Carvone |
0.22 |
0.61 |
alpha-Humulene |
0.04 |
0.02 |
|
Linalyl acetate |
0.12 |
0.16 |
Viridiflorene |
0.08 |
0.001 |
|
Carvenone |
0.03 |
0.02 |
beta-Bisabolene |
1.51 |
2.65 |
|
Thymol |
1.82 |
3.62 |
delta-Amorphene |
0.08 |
0.01 |
|
Carvacrol |
21.29 |
29.22 |
delta-Cadinene |
0.04 |
0.01 |
|
Methyl non-8-enoate |
0.22 |
0.01 |
beta-Sesquiphellandrene |
0.02 |
0.001 |
|
Oregano Sesquiterpenoid 1 |
0.06 |
0.05 |
trans-alpha-Bisabolene |
0.01 |
2.65 |
|
Carvacrol acetate |
0.01 |
0.2 |
Spathulenol |
0.1 |
0.46 |
|
Caryophyllene oxide |
0.2 |
1.01 |
alpha-Cadinol |
0.02 |
0.001 |
|
alpha-Muurolol |
0.09 |
0.07 |
5-trans-9-trans-Farnesyl acetone |
0.05 |
0.01 |
Please see below for detailed comments.
Comment 2:
Title: The title can be improved. A good title is short, concise and informative.
Answer 2: Thank you for the suggestion. We modified the title in „ The antimicrobial activity of Origanum vulgare L. correlated with the gastro-intestinal perturbation in patients with Metabolic Syndrome”
Comment 3:
Abstract: Do not add references in the Abstract. Move the cited reference in the Introduction.
Answer 3: Thank you for the remark. The authors corrected the Abstract, and moved the cited in the introduction.
Comment 4:
Lines 22-25: Please, rephrase it.
Answer 4: Thank you for the amendment. The authors corrected the sentence accordingly. Please, see the correction written with red in the manuscript (lines 22-25).
Comment 5:
Line 26-28: Please use the "italic" for latin name as "Origanum vulgare", Staphylococcus aureus, Escherichia coli and Streptococcus pyogenes. Please, apply this change all over the paper.
Answer 5: Thank you very much for the correction. We modified the paragraph accordingly (lines 26-28), and in all over the paper.
Comment 6:
Line 25-29: This paragraph is not clear. Please rephrase it. Try to split in more sentences.
Answer 6: Thank you for the amendment. The authors corrected the sentence accordingly. Please, see the correction written in red in the manuscript (lines 25-29).
Comment 7:
Line 35: Add a dot after dysbiosis and start a new sentence. Rephrase "in patients also improving the phase angle, used as an index of health and cellular function".
Answer 7: Thank you very much for the correction. We modified the paragraph accordingly (lines 32-33).
Keywords: Do not use as keywords the same words that are already present in the Title, such as Origanum vulgare, dysbiosis and essential oil.
Comment 8: Thank you for the remark. The authors corrected the keywords (lines 34-35)
Results:
Lines 93-95: please delete the double comma.
Answer 8: Thank you very much for the correction. We deteted the double comma (lines 94-96)
Comment 9:
Figure 2: Please describe better the legend of the figure. What are the disks indicated in the figure 2? Antibiotics? EOO? Control?
Answer 9: Thank you for the suggestion. The whole manuscript was extensively revised and corrections were made throughout (lines 99-103).
Comment 10:
Table 1: What does mean a diameter higher than 33? Can also be 100? Please indicate the precise diameters. In the table change the word “Oregano” with “Origan”
Answer 10: Thank you very much for the correction. We made the required corrections for the diameters (lines 104-117), and in the table 1, too.
Comment 11:
Line 131: Please delete the capital letter “The”
Answer 11: Thank you for the remark. The authors corrected the sentence accordingly (lines 141-143).
Comment 12:
Figure 3: this figure is not well presented. Please add axis title, for example what numbers represent. Remove the Title of the figure and try to add more information in the description of the figure.
Answer 12: Gratefully accepting the observation, the paragraph and the legend of Figure 3 were rephrased, was removed the title of the figure, and the explanation of the used was included in the paragraph to be more accurate (lines 133-145 and 147-151).
Comment 13:
Line 139: please change the word “variabile” with "variables".
Answer 13: Thank you very much for the correction. We modified the paragraph accordingly (line 155).
Materials and methods:
Comment 14:
Whic solvent did you use for the extraction? What is the ratio between solvent and organic matter? How much solvent and how much Origanum vulgare did you use?
Line 288: Please add a reference here.
Answer 14: Thank you for the remark. The volatile oil was obtained by the hydrodistillation method. The solvent used was water. From 25 g of dried plant we obtained 4-5 ml of essential oil. This method was detailed in lines 297-302.
Comment 15:
Line 298: Please rephrase the sentence “the culture medium is agar is a solid medium Mueller Hinton…”
Answer 15: Thank you for the amendment. The authors corrected the sentence accordingly. Please, see the correction written in red in the manuscript (lines 326-337).
Comment 16:
Line 306: please rephrase, there is no verb in the sentence, there is no end, there is no subject. This is not academic English.
Answer 16: Thank you for the remark. Accepting the suggestion, we made the required corrections (line 337).
Comment 17:
Line 316: correct the ampicillin (10
Answer 17: Thank you very much for the correction. We modified the paragraph accordingly (line 347).
Comment 18:
Line 325: Rename the paragraph for example - Orgnaization of the study groups. - Clinical trials
Answer 18: Again, we totally agree with your suggestion, and corrections were made correspondingly (line 358)
Comment 19:
Lines 329-331: Please substitute with:
- L1 = patients with Staphylococcus infection (n=32)
- L2 = patients with Escherichia coli infection (n=51)
- L3 = patients with Streptococcus infection (n=2).
Answer 19: Thank you very much for the suggestion, the lines 362-365 was modified accordingly.
Comment 20:
How did you do statistic in L3 group with only 2 patients?
Answer 20: Thank you very much for the comment. The statistical study was performed on 106 patients, of which only two had Steptococcus pyogenes infection. The statistical processing was done as for the study as a whole, from where we extracted the results. We want to continue the study for a longer time and with more patients. Then the statistical calculations will also include several cases.
Comment 21:
Line 358: in "their" hands.
Answer 21: Thank you for the remark. The paragraph was modified accordingly (line 398).
Comment 22:
Line 366: Please rephrase the sentence.
Answer 22: Thank you very much for the correction. We modified the paragraph accordingly (line 408).
Comment 23:
Line 377: Gastrointestinal pain WAS EVALUATED....
Answer 23: Thank you for the amendment. The authors corrected the sentence accordingly (lines 425-427).Gastrointestinal pain was followed with adapted scales (visual analogue scale (VAS), numerical rating scale (NRS), and verbal rating scale (VRS).
Response to Reviewer 1
Firstly, we, the authors of the present manuscript wish to thank you for thoughtful commentary you have provided to improve the quality of the paper. We are very grateful for the time and effort you have devoted to this task. We have extensively revised our manuscript according to the recommendations. All changes in the text and the new figures that we have redesigned are highlighted. Please, see the point-by-point answers to your comments below. The whole manuscript was extensively revised and corrections were made throughout.
Comment 1:
The paper entitled "Correlation of treatment with the essential oil of Origanum vulgare L. of minor bacterial infections with the incidence of dysbiosis in patients with Metabolic Syndrome" showed the antibacterial activity of essential oil of Origanum vulgare L. Overall, the study presented by the authors has good potential. However, an extensive editing of English language is necessary. Authors do not use academic English and in many paragraphs is very difficult to understand what they want to communicate. Authors used “essential oil” of Origanum vulgare (EOO) in their experiment. The essential oils are normally composed by lipophile compounds. However, authors obtained the EOO by hydro distillation in Soxhlet. Did authors used only water or other organic solvent? If they used only water, did they observe phases separation after extraction? If yes, how did they manage the right amount for oral administration in patients? While if authors used organic solvents mixed with water, did they evaluate toxicity before oral administration?
The presented version has lack of information. Results are not presented clearly making the readers difficult to follow even small sentences.
Answer 1
Thank you very much for the observation. The volatile oil was obtained by the hydrodistillation method. There was confusion with the extraction of the fatty oil. The extraction method is detailed between lines 297-302, and we also include the figures of the extraction phases.
The EOO obtained was analyzed with a GC-MS gas chromatograph. The result is shown in Table 3, in line 316. Carvacrol was obtained as the main substance, substance being present 29.22%.
|
- refractive index 1.523 |
Figure 1. Hydrodistillation extraction method of EOO
Table 3. Gas chromatograph report for standard and extract of EOO in percentage composition
|
Compound name |
Standard area % |
Extract area % |
Compound name |
Standard area % |
Extract area % |
|
Methyl-alpha-methyl butyrate |
0.05 |
0.06 |
trans-beta-O cymene |
0.05 |
0.18 |
|
alpha-Thujene |
0.62 |
0.52 |
gamma-Terpinene |
4.02 |
1.06 |
|
alpha-Pinene |
0.44 |
0.23 |
cis-Sabinene hydrate |
0.26 |
0.16 |
|
Thujadiene |
0.01 |
|
Terpinolene |
0.11 |
0.09 |
|
Camphene |
0.15 |
0.01 |
T-Linalool oxide (furanoid) |
0.03 |
0.16 |
|
1-Octen-3-ol |
0.26 |
0.77 |
para-Cymenene |
0.02 |
0.18 |
|
Myrcene |
1.34 |
|
Linalool |
8.99 |
11.18 |
|
3-Octanol |
0.01 |
0.19 |
trans-Sabinene hydrate |
0.04 |
0.01 |
|
alpha-Phellandrene |
0.17 |
0.02 |
Hotrienol |
0.13 |
0.01 |
|
delta-3-Carene |
0.06 |
0.03 |
cis-para-Menth-2-en-1-ol |
0.04 |
0.02 |
|
alpha-Terpinene |
0.94 |
1.09 |
alpha-Campholenal |
0.03 |
0.01 |
|
para-Cymene |
2.92 |
0.18 |
trans-Pinocarveol |
0.02 |
0.02 |
|
Limonene |
0.17 |
1.12 |
trans-para-Menth-2-en-1-ol |
0.03 |
2.89 |
|
beta-Phellandrene |
0.18 |
0.25 |
Borneol |
0.7 |
3.39 |
|
1,8-Cineole |
0.06 |
0.02 |
Terpinen-4-ol |
0.69 |
1.06 |
|
cis-beta-Ocimene |
0.03 |
0.01 |
para-Cymen-8-ol |
0.03 |
0.18 |
|
alpha-Terpineol |
0.25 |
1.06 |
alpha-Bourbonene |
0.02 |
0.00 |
|
cis-Dihydro carvone |
0.03 |
0.01 |
beta-Caryophyllene |
0.93 |
1.44 |
|
trans-Chrysanthenyl acetate |
0.02 |
0.01 |
trans-alpha-Bergamotene |
0.02 |
0.01 |
|
Carvacrol Methyl ether |
0.05 |
|
Aromadendrene |
0.18 |
0.01 |
|
Carvone |
0.22 |
0.61 |
alpha-Humulene |
0.04 |
0.02 |
|
Linalyl acetate |
0.12 |
0.16 |
Viridiflorene |
0.08 |
0.001 |
|
Carvenone |
0.03 |
0.02 |
beta-Bisabolene |
1.51 |
2.65 |
|
Thymol |
1.82 |
3.62 |
delta-Amorphene |
0.08 |
0.01 |
|
Carvacrol |
21.29 |
29.22 |
delta-Cadinene |
0.04 |
0.01 |
|
Methyl non-8-enoate |
0.22 |
0.01 |
beta-Sesquiphellandrene |
0.02 |
0.001 |
|
Oregano Sesquiterpenoid 1 |
0.06 |
0.05 |
trans-alpha-Bisabolene |
0.01 |
2.65 |
|
Carvacrol acetate |
0.01 |
0.2 |
Spathulenol |
0.1 |
0.46 |
|
Caryophyllene oxide |
0.2 |
1.01 |
alpha-Cadinol |
0.02 |
0.001 |
|
alpha-Muurolol |
0.09 |
0.07 |
5-trans-9-trans-Farnesyl acetone |
0.05 |
0.01 |
Please see below for detailed comments.
Comment 2:
Title: The title can be improved. A good title is short, concise and informative.
Answer 2: Thank you for the suggestion. We modified the title in „ The antimicrobial activity of Origanum vulgare L. correlated with the gastro-intestinal perturbation in patients with Metabolic Syndrome”
Comment 3:
Abstract: Do not add references in the Abstract. Move the cited reference in the Introduction.
Answer 3: Thank you for the remark. The authors corrected the Abstract, and moved the cited in the introduction.
Comment 4:
Lines 22-25: Please, rephrase it.
Answer 4: Thank you for the amendment. The authors corrected the sentence accordingly. Please, see the correction written with red in the manuscript (lines 22-25).
Comment 5:
Line 26-28: Please use the "italic" for latin name as "Origanum vulgare", Staphylococcus aureus, Escherichia coli and Streptococcus pyogenes. Please, apply this change all over the paper.
Answer 5: Thank you very much for the correction. We modified the paragraph accordingly (lines 26-28), and in all over the paper.
Comment 6:
Line 25-29: This paragraph is not clear. Please rephrase it. Try to split in more sentences.
Answer 6: Thank you for the amendment. The authors corrected the sentence accordingly. Please, see the correction written in red in the manuscript (lines 25-29).
Comment 7:
Line 35: Add a dot after dysbiosis and start a new sentence. Rephrase "in patients also improving the phase angle, used as an index of health and cellular function".
Answer 7: Thank you very much for the correction. We modified the paragraph accordingly (lines 32-33).
Keywords: Do not use as keywords the same words that are already present in the Title, such as Origanum vulgare, dysbiosis and essential oil.
Comment 8: Thank you for the remark. The authors corrected the keywords (lines 34-35)
Results:
Lines 93-95: please delete the double comma.
Answer 8: Thank you very much for the correction. We deteted the double comma (lines 94-96)
Comment 9:
Figure 2: Please describe better the legend of the figure. What are the disks indicated in the figure 2? Antibiotics? EOO? Control?
Answer 9: Thank you for the suggestion. The whole manuscript was extensively revised and corrections were made throughout (lines 99-103).
Comment 10:
Table 1: What does mean a diameter higher than 33? Can also be 100? Please indicate the precise diameters. In the table change the word “Oregano” with “Origan”
Answer 10: Thank you very much for the correction. We made the required corrections for the diameters (lines 104-117), and in the table 1, too.
Comment 11:
Line 131: Please delete the capital letter “The”
Answer 11: Thank you for the remark. The authors corrected the sentence accordingly (lines 141-143).
Comment 12:
Figure 3: this figure is not well presented. Please add axis title, for example what numbers represent. Remove the Title of the figure and try to add more information in the description of the figure.
Answer 12: Gratefully accepting the observation, the paragraph and the legend of Figure 3 were rephrased, was removed the title of the figure, and the explanation of the used was included in the paragraph to be more accurate (lines 133-145 and 147-151).
Comment 13:
Line 139: please change the word “variabile” with "variables".
Answer 13: Thank you very much for the correction. We modified the paragraph accordingly (line 155).
Materials and methods:
Comment 14:
Whic solvent did you use for the extraction? What is the ratio between solvent and organic matter? How much solvent and how much Origanum vulgare did you use?
Line 288: Please add a reference here.
Answer 14: Thank you for the remark. The volatile oil was obtained by the hydrodistillation method. The solvent used was water. From 25 g of dried plant we obtained 4-5 ml of essential oil. This method was detailed in lines 297-302.
Comment 15:
Line 298: Please rephrase the sentence “the culture medium is agar is a solid medium Mueller Hinton…”
Answer 15: Thank you for the amendment. The authors corrected the sentence accordingly. Please, see the correction written in red in the manuscript (lines 326-337).
Comment 16:
Line 306: please rephrase, there is no verb in the sentence, there is no end, there is no subject. This is not academic English.
Answer 16: Thank you for the remark. Accepting the suggestion, we made the required corrections (line 337).
Comment 17:
Line 316: correct the ampicillin (10
Answer 17: Thank you very much for the correction. We modified the paragraph accordingly (line 347).
Comment 18:
Line 325: Rename the paragraph for example - Orgnaization of the study groups. - Clinical trials
Answer 18: Again, we totally agree with your suggestion, and corrections were made correspondingly (line 358)
Comment 19:
Lines 329-331: Please substitute with:
- L1 = patients with Staphylococcus infection (n=32)
- L2 = patients with Escherichia coli infection (n=51)
- L3 = patients with Streptococcus infection (n=2).
Answer 19: Thank you very much for the suggestion, the lines 362-365 was modified accordingly.
Comment 20:
How did you do statistic in L3 group with only 2 patients?
Answer 20: Thank you very much for the comment. The statistical study was performed on 106 patients, of which only two had Steptococcus pyogenes infection. The statistical processing was done as for the study as a whole, from where we extracted the results. We want to continue the study for a longer time and with more patients. Then the statistical calculations will also include several cases.
Comment 21:
Line 358: in "their" hands.
Answer 21: Thank you for the remark. The paragraph was modified accordingly (line 398).
Comment 22:
Line 366: Please rephrase the sentence.
Answer 22: Thank you very much for the correction. We modified the paragraph accordingly (line 408).
Comment 23:
Line 377: Gastrointestinal pain WAS EVALUATED....
Answer 23: Thank you for the amendment. The authors corrected the sentence accordingly (lines 425-427).Gastrointestinal pain was followed with adapted scales (visual analogue scale (VAS), numerical rating scale (NRS), and verbal rating scale (VRS).
Reviewer 2 Report
The work proposed by Ghitea and colleagues titled:"Correlation of treatment with the essential oil of Origanum vulgare L. of minor bacterial infections with the incidence of diysbiosis in patients with metabolic syndrome", is interesting but fundamental contents are missing. Before considering this paper suitable for publication on Molecules it is necessary that the authors provide the following reviews:
-line 129: check the content of the sentence ..... "and in this case." The sentence does not end correctly.
-The authors chose the essential oil of oregano obtained by the Soxhelet method, as the matrix of their investigation reporting in the discussion (line 233) that carvacrol is one of the major components. But no chemical composition is reported in the manuscript ... I therefore deduce that this statement is consequent to the data reported in the literature.
This is not enough. The authors have to report the percentage chemical composition of the essential oil of oregano they used. Consequently, they must insert a paragraph concerning the method of analysis used (GC / MS) and a table of results must be added.
-Steam distillation is better method to obtain Essential oil . Why the authors didn't they choose this method?
Author Response
Response to Reviewer 2
Firstly, we, the authors of the present manuscript wish to thank you for thoughtful commentary you have provided to improve the quality of the paper. We are very grateful for the time and effort you have devoted to this task. We have extensively revised our manuscript according to the recommendations. All changes in the text and the new figures that we have redesigned are highlighted. Please, see the point-by-point answers to your comments below.
The work proposed by Ghitea and colleagues titled:"Correlation of treatment with the essential oil of Origanum vulgare L. of minor bacterial infections with the incidence of diysbiosis in patients with metabolic syndrome", is interesting but fundamental contents are missing. Before considering this paper suitable for publication on Molecules it is necessary that the authors provide the following reviews:
Comment 1.:
-line 129: check the content of the sentence ..... "and in this case." The sentence does not end correctly.
Answer 1.:
Thank you for the amendment. The authors corrected the sentence accordingly. Please, see the correction highlighted in the manuscript (line 136).
Comment 2:
-The authors chose the essential oil of oregano obtained by the Soxhelet method, as the matrix of their investigation reporting in the discussion (line 233) that carvacrol is one of the major components. But no chemical composition is reported in the manuscript ... I therefore deduce that this statement is consequent to the data reported in the literature.
This is not enough. The authors have to report the percentage chemical composition of the essential oil of oregano they used. Consequently, they must insert a paragraph concerning the method of analysis used (GC / MS) and a table of results must be added. -Steam distillation is better method to obtain Essential oil. Why the authors didn't they choose this method?
Answer 2:
Thank you very much for the observation. The volatile oil was obtained by the hydro-distillation method. There was confusion with the extraction of the fatty oil. The extraction method is detailed between lines 297-302, and we also include the figures of the extraction phases.
The EOO obtained was analyzed with a GC-MS gas chromatograph. The result is shown in Table 3, in line 316. Carvacrol was obtained as the main substance, substance being present 29.22%.
|
|
|
Table 3. Gas chromatograph report for standard and extract of EOO in percentage composition
|
Compound name |
Standard area % |
Extract area % |
Compound name |
Standard area % |
Extract area % |
|
Methyl-alpha-methyl butyrate |
0.05 |
0.06 |
trans-beta-O cymene |
0.05 |
0.18 |
|
alpha-Thujene |
0.62 |
0.52 |
gamma-Terpinene |
4.02 |
1.06 |
|
alpha-Pinene |
0.44 |
0.23 |
cis-Sabinene hydrate |
0.26 |
0.16 |
|
Thujadiene |
0.01 |
|
Terpinolene |
0.11 |
0.09 |
|
Camphene |
0.15 |
0.01 |
T-Linalool oxide (furanoid) |
0.03 |
0.16 |
|
1-Octen-3-ol |
0.26 |
0.77 |
para-Cymenene |
0.02 |
0.18 |
|
Myrcene |
1.34 |
|
Linalool |
8.99 |
11.18 |
|
3-Octanol |
0.01 |
0.19 |
trans-Sabinene hydrate |
0.04 |
0.01 |
|
alpha-Phellandrene |
0.17 |
0.02 |
Hotrienol |
0.13 |
0.01 |
|
delta-3-Carene |
0.06 |
0.03 |
cis-para-Menth-2-en-1-ol |
0.04 |
0.02 |
|
alpha-Terpinene |
0.94 |
1.09 |
alpha-Campholenal |
0.03 |
0.01 |
|
para-Cymene |
2.92 |
0.18 |
trans-Pinocarveol |
0.02 |
0.02 |
|
Limonene |
0.17 |
1.12 |
trans-para-Menth-2-en-1-ol |
0.03 |
2.89 |
|
beta-Phellandrene |
0.18 |
0.25 |
Borneol |
0.7 |
3.39 |
|
1,8-Cineole |
0.06 |
0.02 |
Terpinen-4-ol |
0.69 |
1.06 |
|
cis-beta-Ocimene |
0.03 |
0.01 |
para-Cymen-8-ol |
0.03 |
0.18 |
|
alpha-Terpineol |
0.25 |
1.06 |
alpha-Bourbonene |
0.02 |
0.00 |
|
cis-Dihydro carvone |
0.03 |
0.01 |
beta-Caryophyllene |
0.93 |
1.44 |
|
trans-Chrysanthenyl acetate |
0.02 |
0.01 |
trans-alpha-Bergamotene |
0.02 |
0.01 |
|
Carvacrol Methyl ether |
0.05 |
|
Aromadendrene |
0.18 |
0.01 |
|
Carvone |
0.22 |
0.61 |
alpha-Humulene |
0.04 |
0.02 |
|
Linalyl acetate |
0.12 |
0.16 |
Viridiflorene |
0.08 |
0.001 |
|
Carvenone |
0.03 |
0.02 |
beta-Bisabolene |
1.51 |
2.65 |
|
Thymol |
1.82 |
3.62 |
delta-Amorphene |
0.08 |
0.01 |
|
Carvacrol |
21.29 |
29.22 |
delta-Cadinene |
0.04 |
0.01 |
|
Methyl non-8-enoate |
0.22 |
0.01 |
beta-Sesquiphellandrene |
0.02 |
0.001 |
|
Oregano Sesquiterpenoid 1 |
0.06 |
0.05 |
trans-alpha-Bisabolene |
0.01 |
2.65 |
|
Carvacrol acetate |
0.01 |
0.2 |
Spathulenol |
0.1 |
0.46 |
|
Caryophyllene oxide |
0.2 |
1.01 |
alpha-Cadinol |
0.02 |
0.001 |
|
alpha-Muurolol |
0.09 |
0.07 |
5-trans-9-trans-Farnesyl acetone |
0.05 |
0.01 |
Round 2
Reviewer 1 Report
Although authors made extensive improvement to the paper, other modifications are needed, including english changes.
Abstract:
Line 27 : please write E. coli in italic and with no abbreviation in the abstract "Escherichia coli"
Introduction:
Line 38-40: this sentence is too long, please split it in two to make it more clear.
Line 49-50: the word "which" is repetitive.
Materials and methods:
Authors should add a paragraph about GC-MS experiments in which they explain the insturment they used, the column used, the programm temperature, the type of run (split or splitless) etc....
Authors should add a section in materials and methods in which they explain which standard they used and where they bough it.
Results:
Table 3: How authors calculate the % of each compound? Please explain. Is the standard area the area of standard compounds or something else? Please make it clear.
Table 3: authors should include retention time and m/z of each compound
Line 427: Please substitute the word "followed" with "evaluated". You can't follow a gastrointestinal pain but you can evaluate it
Author Response
Response to Reviewer 1
We are very grateful for the effort and time you have devoted to this task. We, the authors of the present manuscript wish to thank you for thoughtful commentary you have provided to improve the quality of the paper. We have extensively revised our manuscript according to the recommendations. All changes in the text and the new figures that we have redesigned are highlighted. Please, see the point-by-point answers to your comments below.
Comment 1:
Abstract:
Line 27 : please write E. coli in italic and with no abbreviation in the abstract "Escherichia coli"
Answer 1: Thank you very much for the correction. We modified the paragraph accordingly (line 28).
Comment 2:
Introduction:
Line 38-40: this sentence is too long, please split it in two to make it more clear.
Answer 2: Thank you for the suggestion. The whole manuscript was extensively revised and corrections were made throughout. (lines 38-40)
Comment 3:
Line 49-50: the word "which" is repetitive.
Answer 3: Thank you very much for the correction. We modified the paragraph accordingly (lines 49-50).
Comment 4:
Materials and methods:
Authors should add a paragraph about GC-MS experiments in which they explain the insturment they used, the column used, the programm temperature, the type of run (split or splitless) etc....
Authors should add a section in materials and methods in which they explain which standard they used and where they bough it.
Results:
Table 3: How authors calculate the % of each compound? Please explain. Is the standard area the area of standard compounds or something else? Please make it clear.
Table 3: authors should include retention time and m/z of each compound
Answer 4: Thank you for the amendment. The authors corrected the sentences and the table three accordingly. Please, see the correction highlighted in the manuscript.
The ability and high resolution to provide precise and accurate qualitative and quantitative data established gas-chromatography (GC) coupled with mass spectrometry (MS), i.e., GC-MS analyses as a valuable means for taxonomic research of plants. The relative amounts of individual components of the volatiles of the two samples were expressed as percentages of the peak area relative to the total peak area. Relative percentage amounts were calculated from the TIC by the computer. (lines 309-314)
The description of the instrumentation used (model and make) for the GC / MS analysis the authors completed in lines 320-329.
Table three was corrected in line 336
Table 3. Gas chromatograph report for standard and extract of EOO in percentage composition
|
Compound name |
Retention time |
Relative area % |
Compound name |
Retention time |
Relative area % |
|
Standard |
Extract |
||||
|
3,6,6-trimetil-2-norpinene |
6.078 |
1.54 |
α-Pinene |
6.074 |
0.23 |
|
Bicyclo[2.2.1]heptan, 2,2-dimethil-3-methilene-, (lS)- |
6.323 |
0.56 |
Bicyclo[2.2.1]heptan, 2,2-dimethil-3-methilene-, (1S) - |
6.323 |
0.16 |
|
1-Octen-3-ol |
6.714 |
0.69 |
1-Octen-3-ol |
6.710 |
0.77 |
|
Bicyiclo[3,l.1 ]heptan, 6,6-dimethil-2methilene-, IS)- |
6.772 |
0.25 |
3-Octanol |
6.969 |
0.19 |
|
β-Mircene |
6.925 |
3.53 |
o-Cimene |
7.489 |
0.44 |
|
3-Octanol |
6.972 |
0.19 |
Eucaliptol |
7.622 |
0.40 |
|
α-Phellandrene |
7.183 |
0.66 |
2-Furanmetanol, 5-eteniltetrahydro- α,α,5-trimethil-, cis- |
8.248 |
0.23 |
|
(IS)-2,6,6-Trimethilbicyclo [3.1.1]hept 2-ene |
7.282 |
0.23 |
cis-Linaloloxid |
8.496 |
0.16 |
|
(+)-2-Carene |
7.377 |
2.74 |
Linalool |
8.659 |
11.18 |
|
o-Cimen |
7.513 |
5.52 |
endo-Borneol |
9.737 |
3.39 |
|
D-Limonene |
7.574 |
1.12 |
R-(-)-p-Menth-l-en-4-ol |
9.890 |
2.89 |
|
1,3,7-Octatrien, 3,7-dimethil- |
7.823 |
0.24 |
Terpineol |
10.210 |
1.06 |
|
2-Thujene |
8.040 |
6.24 |
p-Cimen-2-ol-methil-ether |
10.790 |
0.18 |
|
β-cis-Terpineol |
8.180 |
0.41 |
Carvona |
11.118 |
0.61 |
|
cis-Linaloloxid |
8.248 |
0.12 |
Thymol |
11.445 |
3.62 |
|
1,4(8)-diee-para-mentha |
8.510 |
0.63 |
Carvacrol |
11.670 |
29.22 |
|
Linalool |
8.663 |
10.18 |
2-Methil-5-(propan-2-yliden)cyclohexan-1,4-diol |
11.877 |
0.98 |
|
Hortineol |
8.710 |
0.44 |
4,4-Dimethilpent-2-enal |
11.969 |
0.19 |
|
endo-Borneol |
9.734 |
1.48 |
o-lzopropilphenethol |
12.033 |
0.27 |
|
(R)-(-)-p-Menth-l-en-4-ol |
9.890 |
2.02 |
(R)-lavandulil acetate |
12.414 |
0.17 |
|
4-tertbutil-o-crezol |
10.790 |
0.33 |
Phenol, 2-methil-5-(1-methiletil)-, acetate |
12.574 |
0.32 |
|
Carvona |
11.098 |
0.42 |
Geranil acetate |
12.656 |
0.23 |
|
Thymol |
11.438 |
2.59 |
Cariophilene |
13.356 |
1.44 |
|
Carvacrol |
11.620 |
21.29 |
2,7-Dimethiloctadiin-3,5-diol-2,7 |
13.734 |
0.52 |
|
2-Methil-5-(propan-2 yliden)cyclohexan-1,4-diol |
11.863 |
0.53 |
β-Bisabolene |
14.349 |
2.65 |
|
Cariophilene |
13.356 |
1.94 |
(-)-Spatulenol |
15.302 |
0.46 |
|
β-Bisabolene |
14.353 |
3.15 |
Cariophilen oxide |
15.397 |
1.01 |
|
(-)-Spatulenol |
15.298 |
0.22 |
.tau.-Cadinol |
15.989 |
0.57 |
|
Cariophilen oxide |
15.393 |
0.41 |
α-Cadinol |
15.997 |
0.02 |
|
Epizonarene |
15.989 |
0.30 |
3-Trifluoroacetate ester pregnenolone |
21.107 |
0.46 |
|
Decalin, I-metoximethil- |
19.016 |
0.30 |
S-lndacen-l-(2H)-ona, 3,5,6,7-tetrahydro-3,3,5,5-tetramethil-8-(3-methilbutil) |
21.508 |
0.20 |
Comment 5:
Line 427: Please substitute the word "followed" with "evaluated". You can't follow a gastrointestinal pain but you can evaluate it
Answer 5:
Thank you for the remark. Accepting the suggestion, we made the required corrections (line 445).
Reviewer 2 Report
The authors have included the part relating to the chemical analysis and the description of the percentage composition of the essential oil under study.
However, I would like to recommend a description of the instrumentation used (model and make) for the GC / MS analysis.
Even more important is the inclusion of the (calculated) retention indices necessary for an accurate identification of the volatile components.
Author Response
Response to Reviewer 2
We are very grateful for the effort and time you have devoted to this task. We, the authors of the present manuscript wish to thank you for thoughtful commentary you have provided to improve the quality of the paper. We have extensively revised our manuscript according to the recommendations. All changes in the text and the new figures that we have redesigned are highlighted. Please, see the point-by-point answers to your comments below.
Comment 1:
The authors have included the part relating to the chemical analysis and the description of the percentage composition of the essential oil under study.
However, I would like to recommend a description of the instrumentation used (model and make) for the GC / MS analysis.
Even more important is the inclusion of the (calculated) retention indices necessary for an accurate identification of the volatile components.
Answer 1:
Thank you for the amendment. The authors corrected the sentences and the table three accordingly. Please, see the correction highlighted in the manuscript.
The ability and high resolution to provide precise and accurate qualitative and quantitative data established gas-chromatography (GC) coupled with mass spectrometry (MS), i.e., GC-MS analyses as a valuable means for taxonomic research of plants. The relative amounts of individual components of the volatiles of the two samples were expressed as percentages of the peak area relative to the total peak area. Relative percentage amounts were calculated from the TIC by the computer. (lines 309-314)
The description of the instrumentation used (model and make) for the GC / MS analysis the authors completed in lines 320-329.
Table three was corrected in line 336
Table 3. Gas chromatograph report for standard and extract of EOO in percentage composition
|
Compound name |
Retention time |
Relative area % |
Compound name |
Retention time |
Relative area % |
|
Standard |
Extract |
||||
|
3,6,6-trimetil-2-norpinene |
6.078 |
1.54 |
α-Pinene |
6.074 |
0.23 |
|
Bicyclo[2.2.1]heptan, 2,2-dimethil-3-methilene-, (lS)- |
6.323 |
0.56 |
Bicyclo[2.2.1]heptan, 2,2-dimethil-3-methilene-, (1S) - |
6.323 |
0.16 |
|
1-Octen-3-ol |
6.714 |
0.69 |
1-Octen-3-ol |
6.710 |
0.77 |
|
Bicyiclo[3,l.1 ]heptan, 6,6-dimethil-2methilene-, IS)- |
6.772 |
0.25 |
3-Octanol |
6.969 |
0.19 |
|
β-Mircene |
6.925 |
3.53 |
o-Cimene |
7.489 |
0.44 |
|
3-Octanol |
6.972 |
0.19 |
Eucaliptol |
7.622 |
0.40 |
|
α-Phellandrene |
7.183 |
0.66 |
2-Furanmetanol, 5-eteniltetrahydro- α,α,5-trimethil-, cis- |
8.248 |
0.23 |
|
(IS)-2,6,6-Trimethilbicyclo [3.1.1]hept 2-ene |
7.282 |
0.23 |
cis-Linaloloxid |
8.496 |
0.16 |
|
(+)-2-Carene |
7.377 |
2.74 |
Linalool |
8.659 |
11.18 |
|
o-Cimen |
7.513 |
5.52 |
endo-Borneol |
9.737 |
3.39 |
|
D-Limonene |
7.574 |
1.12 |
R-(-)-p-Menth-l-en-4-ol |
9.890 |
2.89 |
|
1,3,7-Octatrien, 3,7-dimethil- |
7.823 |
0.24 |
Terpineol |
10.210 |
1.06 |
|
2-Thujene |
8.040 |
6.24 |
p-Cimen-2-ol-methil-ether |
10.790 |
0.18 |
|
β-cis-Terpineol |
8.180 |
0.41 |
Carvona |
11.118 |
0.61 |
|
cis-Linaloloxid |
8.248 |
0.12 |
Thymol |
11.445 |
3.62 |
|
1,4(8)-diee-para-mentha |
8.510 |
0.63 |
Carvacrol |
11.670 |
29.22 |
|
Linalool |
8.663 |
10.18 |
2-Methil-5-(propan-2-yliden)cyclohexan-1,4-diol |
11.877 |
0.98 |
|
Hortineol |
8.710 |
0.44 |
4,4-Dimethilpent-2-enal |
11.969 |
0.19 |
|
endo-Borneol |
9.734 |
1.48 |
o-lzopropilphenethol |
12.033 |
0.27 |
|
(R)-(-)-p-Menth-l-en-4-ol |
9.890 |
2.02 |
(R)-lavandulil acetate |
12.414 |
0.17 |
|
4-tertbutil-o-crezol |
10.790 |
0.33 |
Phenol, 2-methil-5-(1-methiletil)-, acetate |
12.574 |
0.32 |
|
Carvona |
11.098 |
0.42 |
Geranil acetate |
12.656 |
0.23 |
|
Thymol |
11.438 |
2.59 |
Cariophilene |
13.356 |
1.44 |
|
Carvacrol |
11.620 |
21.29 |
2,7-Dimethiloctadiin-3,5-diol-2,7 |
13.734 |
0.52 |
|
2-Methil-5-(propan-2 yliden)cyclohexan-1,4-diol |
11.863 |
0.53 |
β-Bisabolene |
14.349 |
2.65 |
|
Cariophilene |
13.356 |
1.94 |
(-)-Spatulenol |
15.302 |
0.46 |
|
β-Bisabolene |
14.353 |
3.15 |
Cariophilen oxide |
15.397 |
1.01 |
|
(-)-Spatulenol |
15.298 |
0.22 |
.tau.-Cadinol |
15.989 |
0.57 |
|
Cariophilen oxide |
15.393 |
0.41 |
α-Cadinol |
15.997 |
0.02 |
|
Epizonarene |
15.989 |
0.30 |
3-Trifluoroacetate ester pregnenolone |
21.107 |
0.46 |
|
Decalin, I-metoximethil- |
19.016 |
0.30 |
S-lndacen-l-(2H)-ona, 3,5,6,7-tetrahydro-3,3,5,5-tetramethil-8-(3-methilbutil) |
21.508 |
0.20 |